# Simultaneous UHPLC-MS Quantification of Catechins and Untargeted Metabolomic Profiling for Proof-of-Concept Authenticity Determination of *Maytenus* ssp. Samples

**DOI:** 10.3390/molecules27175520

**Published:** 2022-08-27

**Authors:** Rodolfo Santos Duarte, Elisa Ribeiro Miranda Antunes, Alexandra Christine Helena Frankland Sawaya

**Affiliations:** 1Institute of Biology, State University of Campinas, Campinas 13083-971, SP, Brazil; 2Faculty of Pharmaceutical Science, State University of Campinas, Campinas 13083-871, SP, Brazil

**Keywords:** *Espinheira-santa*, *Maytenus ilicifolia*, *Maytenus aquifolium*, epicatechin, mass spectrometry, chemical profile, metabolomics

## Abstract

Due to the widespread use of *Maytenus ilicifolia* leaves in the treatment of gastric ulcers, herbal medicines derived from such species are distributed by the national health system in Brazil. A related species, *Maytenus aquifolium*, is also used for the same disorders, and both are popularly known as *Espinheira-santa*. Due to their popular use, the quality and efficiency of the herbal medicines derived from these species is an important public health issue. The purpose of this study was to develop and test an analytical method that could quantify the content of catechin and epicatechin in dry *Maytenus* spp. leaves and simultaneously obtain their chemical profile to determine authenticity of the leaf samples. Ultra-high performance liquid chromatography coupled to mass spectrometry (UHPLC-MS) was used to quantify these isomers, in the selected ion monitoring (SIM) mode, while simultaneously analyzing the extract in full-scan mode. This approach was successfully applied to the analysis of commercial and authentic samples of *Maytenus* spp. Fewer than half the samples presented the minimum epicatechin content of 2.8 mg per g of dry leaf mass, as specified in the 6th Brazilian Pharmacopoeia (2019) for *M. ilicifolia*. Furthermore, by using untargeted metabolomics, it was observed that the chemical profile of most the samples was not compatible with *M. ilicifolia* leaves, indicating the need for stricter quality control of this material. The method described herein could be used for this control; moreover, its concept could be adapted and used for an ample variety of medicinal plant products.

## 1. Introduction

*Espinheira-santa* is the popular name for both *Maytenus ilicifolia* Mart ex Reissek and *Maytenus aquifolium* Mart, due to their morphological similarities and traditional use. Both exhibit anti-ulcerogenic and analgesic activities, and the increasing interest in both species resulted in their distribution by the Brazilian Unified Health System (SUS) [1,2].

Since 2007, the SUS has financed the distribution of herbal medicines such as *Espinheira-santa* [2]. Therefore, the quality and efficiency of such products is an important public health issue. However, the official Brazilian norms for the standardization and quality control of these phytomedicines are too vague. Therefore, they do not adequately guarantee the botanical identification and quality of the plant material.

The Brazilian Phytotherapic Memento (2016) considers both *M. ilicifolia* and *M. aquifolium* as equivalent but does not state any requirements for their quality control [3]. More recently, the 6th Brazilian Pharmacopoeia (2019) only cites *M. ilicifolia* and requires a minimum of 2.0% total tannins and 0.28% of epicatechin in the dry leaves [4].

However, epicatechin (the marker compound) and other catechins are also found in several species of *Maytenus* [5] as well as in other plant species that are commonly used to adulterate this herbal medicine [6]. Therefore, even though the analysis of total tannin contents is a simple and cheap laboratory process, it is not adequate for the quality control of these species, as many other plants contain equivalent or even higher total tannin contents [7,8]. In addition, their quantification alone, even by chromatographic methods, does not guarantee the botanical identification of these medicinal plants.

Considering the drawbacks of the present norm and the widespread use and therapeutic importance of *M. ilicifolia* and *M. aquifolium*, an improved method of quality control was deemed necessary. In this regard, associating the use of traditional quantification methods with untargeted metabolomic techniques poses as a good alternative.

Metabolomics is an approach focused on the analysis of the metabolism of organisms and it is usually subdivided into two techniques: targeted and untargeted. Targeted metabolomics aims to detect and identify specific target compounds while untargeted metabolomics focuses on detecting as many compounds as possible, providing a general view of the organism’s metabolism [9].

For both targeted and untargeted metabolomics, the use of high-performance liquid chromatography associated with mass spectrometry (HPLC-MS) is commonly applied due to some useful properties of the method. HPLC-MS enables the separation of thousands of peaks from biological samples [10] in a single run, leading to the detection of a wide range of compounds [11,12]. For this reason, HPLC-MS has increasingly been applied to analysis of herbal medicines [13] and is ideal for the quality control of the plant material as well as the herbal medicine.

Therefore, the purpose of this study was to develop and validate a method using ultra-high performance liquid chromatography coupled to mass spectrometry (UHPLC-MS) with electrospray ionization to separate and quantify the cis/trans isomers (catechin and epicatechin) in the selected ion mode (SIM), while simultaneously acquiring a full scan to perform untargeted metabolomics. As a result, it was possible to monitor epicatechin and catechin content as well as the chemical profile of samples, which is unique to each species, in the same 10 min chromatographic run, adding a second dimension to the analysis.

This method was used for proof-of-concept analysis of authentic samples of *M. ilicifolia* and *M. aquifolium* leaves, as well as hybrid plants and samples of leaves obtained from markets and pharmacies in the State of São Paulo. Although several samples contained the necessary epicatechin content for *M. ilicifolia*, according to the Brazilian Pharmacopoeia (2019) [4], their chromatographic profile told a different story. To the best of our knowledge, this is the first time that targeted and untargeted strategies have been performed simultaneously to obtain complementary information about *Maytenus* spp. sample quality and authenticity.

## 2. Results and Discussion

### 2.1. Analytical Method Validation

Initially, the method using SIM mode to quantify catechin and epicatechin while simultaneously acquiring a full scan was validated according to Brazilian and international legislation [14]. When comparing the UHPLC-MS chromatograms of the standards and solvent (Appendix A), only the standards were detected (catechin at 2.7 min and epicatechin at 3.2 min), confirming the selectivity.

The matrix effect of the leaf extracts on the quantification was evaluated by comparing a series of dilutions of the standards in the solvent or added to the extract. Since the resulting lines were parallel (Appendix A) and the comparison of their angular coefficient resulted in a variation of 1.72% for catechin and 2.5% for epicatechin, the matrix effect on the quantification of these compounds was considered insignificant and all further validation was conducted with the working solution of standards diluted in purified water.

The method was linear for both standards between 0.10 and 2.00 µg/mL. The similarity between analytical curves, Cochran test of homoscedasticity, significant differences between the angular coefficient, linear coefficient, correlation and normality of residues were evaluated; the results are presented in Table 1. As both curves were coincident, that is, the two lines were not significantly different at the intercept and angle (*p* > 0.05), future studies could use an analytical curve of catechin standard (cheaper and easier to find) to quantify both catechin and epicatechin (Appendix A).

Precision was evaluated on two different days to confirm the repeatability and intermediate precision. Six aliquots of a sample were quantified using an external calibration curve built with concentrations of 0.25, 0.50, 1.00, 1.50 and 2.00 µg/mL. The results shown in Table 1 and Appendix A satisfy the norm [14] and the intra- and inter-day’s relative standard deviation was <15%.

Accuracy was evaluated using three concentrations of standards (low 0.1 µg/mL, medium 1.0 µg/mL, high 2.0 µg/mL) and the concentration (recovery) calculated using the linear calibration curves for catechin (y = 291,025x + 8788.6) and epicatechin (y = 298,079x + 32,967). The values presented in Appendix A were acceptable: between 97.90% and 109.33% for all three levels of both standards. According to the AOAC guide [15], acceptable values are between 80% and 110%. Therefore, the results of this method can be considered accurate and precise, as shown in Table 1.

The detection limits (DL) and quantification limits (QL) were determined using triplicate analytical curves; DL for both standards were 0.03 µg/mL and QL for both standards were 0.10 µg/mL (Appendix A). These limits, shown in Table 1, were considered satisfactory since the limit of quantification would be equivalent to 0.25 mg/g of dry leaves, which is 10 times less than the minimum acceptable epicatechin content of 2.8 mg/g of dried leaves [4].

Several parameters of the method were varied to evaluate its robustness, including the use of a different C18 UHPLC column. The extraction was performed using a PDVF 0.22 µm syringe-driven filter or centrifugation for the separation of solids. The flow, column temperature and percentage of formic acid were varied, and compared to the validated method using a solution of catechin and epicatechin (concentration of 1.0 µg/mL). All the results varied within the acceptable range of 95–105% recovery, indicating the method is robust (Appendix A). However, one precaution must be taken: the use of freshly prepared samples and standards. Standard solutions were prepared and left at room temperature for 36 h. The recovery fell from the initial value of 100% to 96.5% and 95.2% for catechin and epicatechin, respectively, after 24 h, and to 89.7% and 88.4% for catechin and epicatechin, respectively, after 36 h.

Therefore, the method was validated and the parameters of linearity, precision, accuracy, repeatability, robustness as well as limits of detection and quantification were satisfactory for the intended use. After validation, we were confident that the SIM mode could be used to quantify catechin and epicatechin while simultaneously acquiring a full scan to evaluate the untargeted profile of the samples.

### 2.2. Quantification and Metabolomics

Using the validated method, the quantification and metabolomics were performed. By analyzing the chromatograms, the peaks of catechin (2.7 min) and epicatechin (3.3 min) could be observed in the SIM (*m*/*z* 289) mode of *M. ilicifolia* (Figure 1. Mi SIM), and in the full scan along other peaks (Figure 1. Mi). In addition, chromatograms of the samples E (Figure 1. E) and *M. aquifolium* (Ma) (Figure 1. Ma) were also similar to Mi, with the exception that the catechin peak (2.7 min) was not present in Ma. Finally, although the full-scan chromatogram of one sample of packaged leaves purchased from a pharmacy (Figure 1. B) presents intense peaks at the beginning of the chromatogram, the peaks of the two catechins are not visible.

According to the Brazilian Pharmacopoeia [4], *M. ilicifolia* should present at least 2.8 mg of epicatechin per g of dry leaves and 2% total tannins. Tannins were not evaluated herein, as previous studies have shown that this parameter is not adequate for the quality control of these species [7,8].

Unfortunately, but not surprisingly, when the validated method was applied to the quantification of catechins in the authentic and commercial leaf samples, more than half of the commercial samples did not present the required content, indicating an urgent need for better quality control of this material. These results are presented in Table 2.

As observed, the authentic samples of *M. ilicifolia* (Mi) and the hybrid (Hb) contained more than 2.8 mg of epicatechin, whereas *M. aquifolium* (Ma) did not. The commercial samples E and F contained enough epicatechin to be considered as *M. ilicifolia*, and brand A presented almost the required amount, with 2.3 mg of epicatechin. Therefore, following the present norm [4], all the samples that contained the acceptable amount of epicatechin could be used.

Simultaneously with the quantification, untargeted metabolomic analysis of the full-scan chromatograms was also performed. Using metabolomic pre-processing techniques, 140 compounds (features) were detected in the samples. These features were identified by their *m*/*z* and retention times. These data were analyzed using ANOVA and post-Hoc Fisher test (*p* < 0.05), indicating that only 78 features were statistically significant. Further analysis was restricted to these features. Using a Random Forest algorithm, 20 features were considered to have the greatest impact on the separation of the samples. This group of features was used to create a heatmap alongside a dendrogram to check the similarity between the samples (Figure 2).

By analyzing the heatmap and dendrogram, it was possible to observe that the samples were separated into two major groups. The authentic *M. ilicifolia* sample was in one group and the *M. aquifolium* and Hybrid samples in another. The heatmap, however, does not group the samples in the same way as the epicatechin content does alone.

Samples E and F, as stated before, presented about 5 mg of epicatechin and therefore could be consumed as *M. ilicifolia*. However, when their total chemical profile was considered, many features presented different intensities to the authentic Mi sample. Therefore, based on their overall composition, samples E and F were classified as similar to *M. aquifolium* and the hybrid (Figure 2). Their epicatechin content could be the result of adulteration or, at best, the samples could belong to another species of the *Maytenus* genus [6,16].

Following the same trend, the sample of the hybrid individual (Hb) would also be approved due to its epicatechin content. However, chemical profile analysis showed the composition of the hybrid samples was clearly distinct from *M. ilicifolia* (Figure 1 and Figure 2) and more similar to *M. aquifolium* (Ma).

By contrast, samples A and C did not present the required amount of epicatechin, but by analyzing the heatmap, it was possible to observe that their overall chemical profile was closer to *M. ilicifolia* (Mi) than to Ma and Hb (Figure 2). Although the samples were within their declared shelf life, it is possible that they were not kept in the correct manner, leading to compound degradation.

Additionally, the epicatechin content of the *M. aquifolium* (Ma) sample was much lower than that of Mi (Table 1), and the difference between species was confirmed by the metabolomic analysis of the full chemical profile. Therefore, contrary to what is described in the Phytotherapic Memento [3], *M. ilicifolia* and *M. aquifolium* are not chemically interchangeable, as has also been stated by Holnik et al. (2015) [17].

Finally, samples B and D presented the lowest amounts of catechin and epicatechin, but they were not grouped together. For sample B, the catechin/epicatechin content was below the limit of quantification, but it was grouped with *M. aquifolium*, although the similarities were not strong. Sample D, on the other hand, presented 0.4 mg and 0.9 mg of catechin and epicatechin respectively per g of dry leaf, but due to the overall profile was grouped with *M. ilicifolia.*

As observed, none of the commercial samples simultaneously presented the same amount of epicatechin and a similar chemical profile as *M. ilicifolia*. The samples presented either the equivalent (or higher) amounts of epicatechin, i.e., quantitative results, or a similar chemical profile, i.e., qualitative results. The quantification of epicatechin alone, therefore, is not enough to determine the quality and authenticity of a sample.

One reason for such results could be due to catechin and epicatechin being ubiquitous compounds present in many plant species. Iacopini et al. (2008), for instance, found similar amounts of catechin and epicatechin to those expected for *M. ilicifolia* in red grape seeds [18]. Furthermore, Ho et al. (1992) detected even larger amounts of epicatechin in Chinese tea samples, ranging from 3% to 8% of sample weight [19], which could easily be added as an adulterant in ground samples.

In addition, it has been demonstrated that the medicinal properties of *Maytenus*, and many other medicinal species, are related to a group of compounds (phytocomplex) rather than a single component [20]. Therefore, the monitoring of entire plant composition would be a better approach to guarantee the quality of a herbal product. The use of metabolomic techniques, as applied herein, has proved to be an interesting approach [21].

In the present work, metabolomics was associated with quantification to develop a better quality-control method for *M. ilicifolia*, using a pool of authenticated individuals as standards for these species. A previous study with *M. ilicifolia*, *M. aquifolium* and hybrid individuals showed that their composition was not strongly affected by the seasons [9]; therefore, the pooled leaves that were used can be considered representative of the composition of these species.

The method developed herein was successfully applied for proof-of-concept analysis of authenticity and quality of samples of *M. ilicifolia* and *M. aquifolium* leaves and is the first time that targeted and untargeted strategies have been performed simultaneously with *Maytenus* spp. samples. Further studies with larger numbers of samples and improved techniques are being undertaken. Moreover, its concept can be adapted and applied to a wide variety of medicinal plant products.

## 3. Materials and Methods

### 3.1. Plant Material

Leaves of *Maytenus ilicifolia* Mart ex Reiss, *Maytenus aquifolium* Mart. and hybrid individuals growing in the Chemical, Biological and Agricultural Pluridisciplinary Research Centre (CPQBA) of the State University of Campinas (UNICAMP) were collected, frozen (−80 °C) and freeze-dried. Leaves from five individuals of each species were collected and pooled together to create each extract.

Vouchers of these specimens were deposited at the UNICAMP Herbarium under the numbers: UEC199156, UEC199157 and UEC199158 (Appendix A), respectively. Samples of leaves sold as *Espinheira-santa* acquired in markets and pharmacies in the State of São Paulo are described in Table 2, although brand names have been omitted. Both authentic plant samples and commercial samples were collected in 2017. As these experiments were performed in 2018, all samples were analyzed before the expiration date on the package (when present), but loose leaves sold in markets did not have an expiration date. Samples were analyzed in triplicate.

### 3.2. Sample Extraction

After testing different proportions of solvents and concentrations of the extract, the optimized procedure for sample extraction was the following: 20 mg of dry leaves in 10 mL of ultrapure water, with extraction in sonic bath, at room temperature for 30 min. This procedure resulted in a recuperation of 97% of catechin and epicatechin. After extraction, the samples were filtered and an aliquot of 1 mL was diluted to a final volume of 5 mL, before analysis by UHPLC-MS. The final dilution was 20 mg of dry leaves in 50 mL of water, much lower than indicated in the Brazilian Phytotherapic Formulary [22] for infusions (3 g/150 mL water), but compatible with UHPLC-MS analysis.

### 3.3. Analytical Method

The analytical method was performed using an Ultra-High Performance Liquid Chromatograph (Acquity, Waters) coupled with a TQD Mass Spectrometer (Acquity, Waters). The analysis was conducted in triplicate. Ionization was performed via electrospray in the negative ion mode (ESI-) with capillary voltage of −4 kV, cone of −25 V, capillary temperature of 150 °C and desolvation temperature of 250 °C. In order to obtain a sensitive and selective method for the quantification of the isomers (catechin and epicatechin) selected ion monitoring (SIM) of *m*/*z* 289 was performed. Due to the rapid switching capacity of quadrupoles, a full scan ranging between *m*/*z* 100 and 1500 (TIC) was acquired simultaneously.

Although the chromatographic conditions were optimized for the separation of the isomers, the gradient elution also permitted a satisfactory separation of the remaining components of the extract. Initially, several solvent systems were tested: purified water with formic acid or ammonium hydroxide, methanol and acetonitrile. The best chromatographic resolution, peak intensity, and the lowest system pressure were obtained using purified water with 0.1% formic acid as solvent A and acetonitrile (HPLC grade, Merck) as solvent B, with a C18 BEH Acquity Waters^®^ (column 1.7 µm × 2.1 mm × 50 mm), flow of 0.2 mL/min, oven temperature of 30 °C and injection volume of 2 µL). The gradient started at 5% B, ramped to 25% in 4.00 min, 50% B in 6.10 min, 95% B in 6.20 min, held until 8.50 min, returned to the initial conditions at 8.51 min and equilibrated the column until 10 min.

The quantitative method was validated for selectivity, linearity, precision, intermediate precision, accuracy and robustness. In addition to the above parameters, the detection limit and quantification limit values were also determined according to Brazilian and international legislation [14]. The identification and quantification of (+/−) catechin (Sigma-Aldrich, 98% purity) and (-) epicatechin (Sigma-Aldrich, >95% purity) were based on comparisons to analytical standards. Stock solutions of the standards were prepared (1.000 mg/mL in methanol), then an aliquot of 10 µL of each was diluted in purified water in a 10 mL flask to an initial concentration of 1 µg/mL (working solution). Dilutions of this working solution in purified water were used for method validation and quantification. Other components cited in Figure 2 were putatively identified in a previous study [9].

### 3.4. Untargeted Processing of Full-Scan Chromatograms

After UHPLC-MS analysis, the raw data were preprocessed using the MarkerLynx tool of the MassLynx software. This step generated a table of features (compounds) that was submitted to statistical analysis using MetaboAnalyst online software [23].

Before statistical and exploratory data analysis, the missing values were processed using KNN estimation based on similar samples (KNN sample-wise), and features with more than 50% of missing values were excluded. The data were also normalized by median and auto-scaled.

With the resulting dataset, statistical analysis of variance (ANOVA) followed by the Fisher post-hoc test (*p* < 0.05) test was performed to determine significant features.

Important features were also extracted using Random Forest algorithm, created with 10 predictors and 500 decision trees, with a constant randomness.

## 4. Conclusions

Herein, we have demonstrated the gain in analytical power by associating metabolomics with traditional chromatographic quantification methods. The association between a targeted analytical method (SIM) for quantification and untargeted metabolomic analysis added a second dimension to the quality control procedures and authenticity of the analyzed samples. The method proposed herein is feasible as it furnishes two distinct but complementary results, requiring no additional sample preparation. Furthermore, the analyses can be carried out simultaneously.

The majority of the commercial leaf samples analyzed in this study did not present the same chemical profile as *M. ilicifolia* or had lower concentrations of epicatechin, indicating the need for stricter quality control of this material. This could be attained by applying the method described herein.

## Figures and Tables

**Figure 1 molecules-27-05520-f001:**
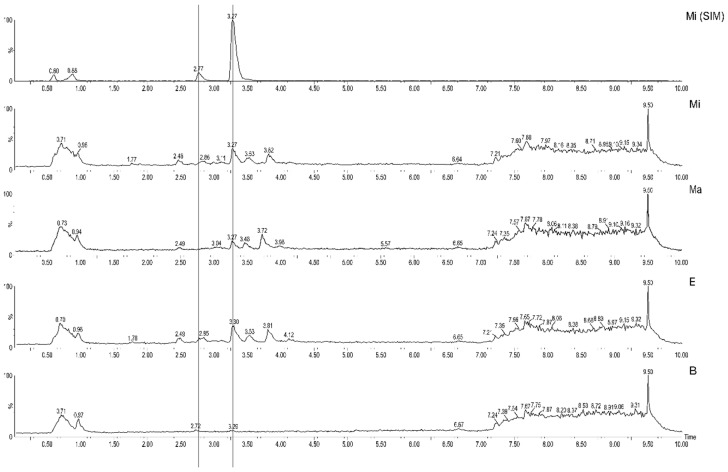
Selected UHPLC-MS chromatograms in SIM mode (MiSIM−*M. ilicifolia*) and full scan (Mi, Ma, E and F) named according to Table 2. Peaks of catechin (2.7 min) and epicatechin (3.3 min) are each marked with a line.

**Figure 2 molecules-27-05520-f002:**
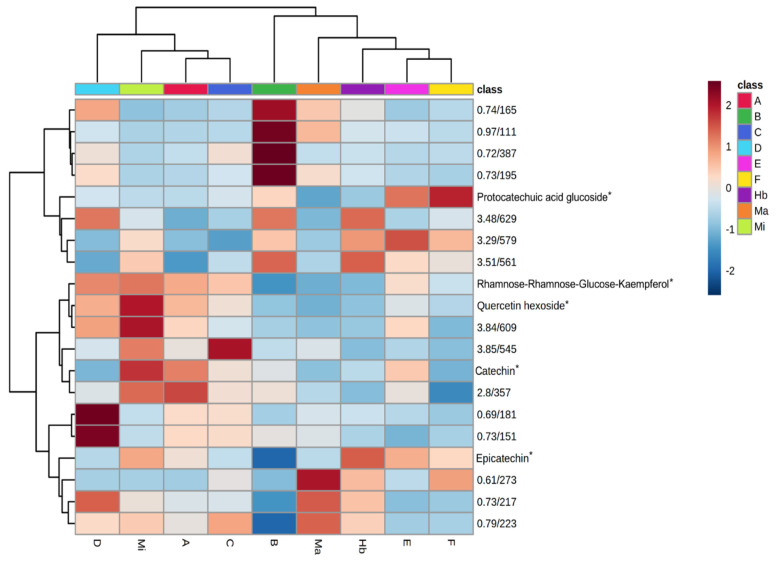
Heatmap showing the 20 most significant features of the UHPLC-MS results, samples identified according to Table 2. Asterisks (*) indicate the compounds identified in a previous study [9].

**Table 1 molecules-27-05520-t001:** Results of the validation and standard parameters of the quantification method for catechin and epicatechin.

Parameter	Standard Values	Calculated Values Catechin	Calculated Values Epicatechin
Homoscedasticity	0.616	0.553	0.427
Correlation Coefficient (*r*)	0.99	0.9971	0.9929
Determination Coefficient (*r*^2^)	-	0.9986	0.9964
Significance of the Angular Coefficient	6.12	5534	2223
Evaluation of the Linear Coefficient (intercept)	4.30	2.00	0.00
Independence of Residues	1.5	1.58	1.57
Normality of Residues	0.908	0.937	0.962
Evaluation of Outliers	3.00	No outliers	No outliers
Variation (%) Angular Coef. between Solvent and Plant Matrix	-	1.72	2.5
Repeatability (RSD%)	<15	6.5	4.0
Intermediate Precision (RSD%)	<15	7.0	5.4
Accuracy (recovery %)	80–110	98.5–109.33	97.90–103.12
Detection Limit (µg/mL)	-	0.03	0.03
Quantification Limit (µg/mL)	-	0.1	0.1

- Not applicable.

**Table 2 molecules-27-05520-t002:** Catechin and epicatechin content of authenticated and commercial dry leaf samples. (<LQ: below quantification limit).

Sample Name	Description	Acquired in	Expiry Date	Catechin (mg/g)	Epicatechin (mg/g)
Mi	*M. ilicifolia*	CPQBA Campinas/SP	-	2.0	4.4
Ma	*M. aquifolium*	CPQBA Campinas/SP	-	<LQ	1.9
Hb	*Maytenus* hybrid	CPQBA Campinas/SP	-	0.8	4.1
A	brand A (packaged)	Pharmacy Itapira/SP	Apr-19	1.0	2.3
B	brand B (packaged)	Pharmacy Itapira/SP	Apr-19	<LQ	<LQ
C	brand C (packaged)	Pharmacy Itapira/SP	Feb-19	0.3	1.1
D	brand D (bulk)	City Market Lindóia/SP	-	0.4	0.9
E	brand E (bulk)	City Market Jacareí/SP	-	1.3	5.5
F	brand F (bulk)	City Market Araraquara/SP	-	0.7	5.2

- not found.

## Data Availability

Not applicable.

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
