# Peer review of "Simultaneous UHPLC-MS Quantification of Catechins and Untargeted Metabolomic Profiling for Proof-of-Concept Authenticity Determination of Maytenus ssp. Samples"

_molecules, 2022, doi:10.3390/molecules27175520_

Round 1
Reviewer 1 Report
General:
The authors developed an analytical method using by UHPLC-MS method to determine catechin and epicatechin in leaf samples to compare with Brazilian Pharmacopoeia as quality control. Furthermore, the authors study the chromatographic profile to determine the authenticity of the leaf samples.
Before starting, it is essential to explain the difference between the two species (Maytenus ilicifolia and Maytenus aquifolium). According to The Plant List, a web database, both species are synonyms. Please provide enough evidence to explain the differences. Or just report as Maytenus spp.
Please, at the first time, mention the species name and provide the complete name with authority, such as:
Maytenus ilicifolia Mart. ex Reissek
Maytenus aquifolium Mart.
What are the difference our your work with the work of Soares et al. (2004) (DOI: https://doi.org/10.1016/j.jpba.2004.08.029) and the work of Souza et al. (2008) (DOI: https://doi.org/10.1016/j.jpba.2007.12.008)? You do not cite these manuscripts in your paper, and it is essential to improve your discussion. Please review these works.
Why select SIM and not Multiple Reaction Monitoring (MRM)? When you only use SIM in plant extract, it is possible to monitor the same ion for other metabolites; however, when you use MRM, you have better selectivity. Please explain this situation and provide evidence that it is a better SIM than MRM. Remember that you aim to develop a quality control method. Before you say that you use this methodology because it is simple and easy, why not use only UV-DAD? It’s easier and economical.
Results and discussion:
Results and discussion:
In my opinion, I think is better to include a summary of validation parameters in the manuscript. Please include these parameters in one or two tables. Thank you.
The author provides enough information to demonstrate the precision and accuracy of their quality control method.
It is essential to improve the discussion and better explication in the part of metabolomics, the heatmap. How and why did you select these ions? How you are entirely sure that these ions are for the sample? These questions were born because the authors use QqQ equipment, which only has low mass spectrometry resolution and the noise in this equipment generally is high. Please, this section is essential to clarify because it is one of the goals of your work. When reading this section is difficult to understand.
Conclusion:
Please, this is a scientific journal with high impact. It is not a diffusion journal; for that, do not conclude with a question or a suggestion focused on the scientific rigor of your results. Please, rewrite your conclusions according to your results and your goals. That's all.
Author Response
Reviewer 1
General:
The authors developed an analytical method using by UHPLC-MS method to determine catechin and epicatechin in leaf samples to compare with Brazilian Pharmacopoeia as quality control. Furthermore, the authors study the chromatographic profile to determine the authenticity of the leaf samples.
Before starting, it is essential to explain the difference between the two species (Maytenus ilicifolia and Maytenus aquifolium).
According to The Plant List, a web database, both species are synonyms. Please provide enough evidence to explain the differences. Or just report as Maytenus spp.
Answer- The status for the classification of M. aquifolium as a synonym of M. ilicifolia is still under review (http://www.theplantlist.org/tpl1.1/record/kew-2370220). According to The Plant List website itself, the ‘WCSP (in review)’ status indicates that the data provided awaits a review by specialists or is still being compiled (http://www.theplantlist.org/1.1/about/#wcsir).
In addition, the website also states “The Plant List is static. It is neither updated regularly from the original data sources, nor edited directly. Data was extracted from source databases in May 2012 and thus records included here may differ from their current equivalent records in the source database from which they were taken. Where you suspect errors in The Plant List, please first check the source databases where corrections may have already been made.”
Upon checking the International Plant Names Index (https://www.ipni.org) and the Kew Botanical Garden’s database, the sources from which the information on Maytenus ilicifolia were obtained, according to The Plant List, it is possible to observe that the Maytenus aquifolium is not considered its synonym.
Maytenus ilicifolia: https://www.ipni.org/n/161937-1 / https://powo.science.kew.org/taxon/161937-1#synonyms
Maytenus aquifolium: https://www.ipni.org/n/161815-1
Finally, many studies have demonstrated the differences between the species, mainly related to their chemical composition and pharmacological effects. The species share many similarities, and it is possible that due to such similarities, some authors consider them as synonyms especially since they can generate a viable hybrid. However, as mentioned, both species have differences in chemical composition and pharmacological activities and to this date, they have not yet been classified as synonyms.
Below, some of studies that demonstrated differences in chemical composition and pharmacological activities between both species :
https://doi.org/10.1590/1983-084X/12_160
https://doi.org/10.1002/ptr.2650080411
https://doi.org/10.1016/j.jchromb.2006.09.014
https://doi.org/10.1016/j.indcrop.2020.113014 (Previous study from our research group)
Please, at the first time, mention the species name and provide the complete name with authority, such as: Maytenus ilicifolia Mart. ex Reissek, Maytenus aquifolium Mart.
Answer- The complete name with authority was already provided in the original manuscript, see lines 33-34
What are the difference our your work with the work of Soares et al. (2004) (DOI: https://doi.org/10.1016/j.jpba.2004.08.029 - and the work of Souza et al. (2008) (DOI: https://doi.org/10.1016/j.jpba.2007.12.008 - You do not cite these manuscripts in your paper, and it is essential to improve your discussion. Please review these works.
Answer- The purpose of this study was to simultaneously quantify catechin/epicatechin and analyze the chemical profiles of the species, as the amount of these chemical markers by itself does not guarantee the identity of the species. While Soares et al. (2004) developed a method for the sole quantification of catechin and epicatechin, using LC-UV, our study also monitored the global chemical profile of the species using UHPLC-MS, which is more sensitive and provides more information of the sample profile. The same goes for Souza et al. (2008), in which the focus was to quantify and identify flavonoids and tannins from the species which was not the objective of the present study.
Why select SIM and not Multiple Reaction Monitoring (MRM)? When you only use SIM in plant extract, it is possible to monitor the same ion for other metabolites; however, when you use MRM, you have better selectivity. Please explain this situation and provide evidence that it is a better SIM than MRM. Remember that you aim to develop a quality control method. Before you say that you use this methodology because it is simple and easy, why not use only UV-DAD? It’s easier and economical.
Answer- We used SIM + full scan rather than MRM + full scan for a technical reason. In the UHPLC-MS equipment used in our lab, the use of collision gas (necessary for MRM) seriously reduces the sensitivity of full scan and would interfere in the proposition of quantifying the catechin/epicatechin content while simultaneously acquiring a full scan to monitor the chemical profile of the species. Furthermore, the selectivity of the SIM quantification and other parameters were fully validated (lines 87-144), demonstrating that, for this study, SIM mode was sufficient. However, in other types of MS equipment, in which MRM + full scan is compatible, this combination could be used.
UV-DAD, could be used for the quantification of the catechin/epicatechin content, which is nothing new (Soares et al., 2004) but would not be able to obtain the chemical profile of the whole sample, which is important for this study.
Results and discussion:
In my opinion, I think is better to include a summary of validation parameters in the manuscript. Please include these parameters in one or two tables. Thank you.
The author provides enough information to demonstrate the precision and accuracy of their quality control method.
Answer -Although the details could be found in the Supplementary material, we have added this information in Table 1.
It is essential to improve the discussion and better explication in the part of metabolomics, the heatmap. How and why did you select these ions? How you are entirely sure that these ions are for the sample? These questions were born because the authors use QqQ equipment, which only has low mass spectrometry resolution and the noise in this equipment generally is high. Please, this section is essential to clarify because it is one of the goals of your work. When reading this section is difficult to understand.
Answer- The traditional metabolomics workflow is described in the material and methods (4.4). In short after chromatograms are obtained, the data in these chromatograms is extracted; one of the parameters is the noise threshold, so eventual contamination from solvents (for example) is removed. As described in lines 179-187, 140 features were observed; these features were identified by their m/z and retention times (this has been added to the text). However not all features were statistically significant; ANOVA and post-Hoc Fisher test (p<0.05); reducing the analysis to 78 features. Using a further algorithm (Random Forest algorithm) the 20 most significant features for the separation of samples were selected for the heatmap.
In summary, the ions obtained by the metabolomics analysis were further selected based on statistical tests of ANOVA and post hoc fisher test alongside with the Random Forest Algorithm attribute of feature importance. With such tests inconsistent ions (that do not present consistent intensity or presence alongside the samples) were filtered out.
Conclusion:
Please, this is a scientific journal with high impact. It is not a diffusion journal; for that, do not conclude with a question or a suggestion focused on the scientific rigor of your results. Please, rewrite your conclusions according to your results and your goals. That's all.
Answer- The conclusion was rewritten.
Reviewer 2 Report
Please check the English form
Author Response
Thanks for your review, we have revised the English and expect to have removed all the mistakes.
Reviewer 3 Report
Submitted manuscript fits the journal's aim and scope and could be of interest of readers.
It can be considered for publication after several corrections:
Don’t forget to write „Maytenus“ in italics anywhere in the text.
Line 16: are known popularly known as Espinheira-santa...
Avoid too long sentences.
Re-write an abstract and point out the most important findings of your research.
Not necessary to mention in the abstract that catechin and epicatechin are isomers. The readers of Molecules definitely know that.
Line 35 and 36: This statement has to be supported by the reference.
There are four catechin diastereoisomers. How these have been recognized?
It has to be clearly designated anywhere in the text which isomers you worked with.
Table 1, I don’t understand what „scheme“ does mean? Explain it, please, or rename it.
All in all, table 1 could be prepared in a more precise manner.
In Table 1, I believe that calculated values are listed for catechin and epicatechin (not only catechin).
When did you collect plant material?
For better orientation, I recommend you to assign Figures and Tables in Supplementary material as S1 and so on. This way assign them also in the text of your manuscript.
Table S5, what does it mean „other C18 column? Did you use different stationary phases or only C18?
Please, provide scans of vouchers of specimens as supporting material. If it is possible, provide also photos of the purchased plant material.
Read the whole manuscript conscientiously, please, and correct several typos and discrepancies.
Author Response
Reviewer 3
Submitted manuscript fits the journal's aim and scope and could be of interest of readers.
It can be considered for publication after several corrections:
Don’t forget to write „Maytenus“ in italics anywhere in the text.
Answer- The manuscript was revised
Line 16: are known popularly known as Espinheira-santa... - Avoid too long sentences.
Answer- The manuscript was revised and long sentences were modified
Re-write an abstract and point out the most important findings of your research. Not necessary to mention in the abstract that catechin and epicatechin are isomers. The readers of Molecules definitely know that.
Answer- We have modified the abstract accordingly.
Line 35 and 36: This statement has to be supported by the reference.
Answer- The phrase (lines 35 -37) was modified and the reference added.
There are four catechin diastereoisomers. How these have been recognized? It has to be clearly designated anywhere in the text which isomers you worked with.
Answer-We used the following standards: (-) Epicatechin - Sigma-Aldrich and (+/-) Catechin – Sigma-Aldrich, this has been added to the text.
Table 1, I don’t understand what „scheme“ does mean? Explain it, please, or rename it.
Answer- couldn’t find the word “scheme” in table 1.
All in all, table 1 could be prepared in a more precise manner. In Table 1, I believe that calculated values are listed for catechin and epicatechin (not only catechin).
Answer- The mistake in Table 1 was corrected, one column is for catechin and the other for epicatechin. More information was included, as another referee suggested.
When did you collect plant material?
Answer-The plant material was collected during 2017as well as the commercial samples. The information was added to the material and methods (4.1).
For better orientation, I recommend you to assign Figures and Tables in Supplementary material as S1 and so on. This way assign them also in the text of your manuscript.
Answer- All supplementary material has bee renamed adding S before the number, as suggested.
Table S5, what does it mean „other C18 column? Did you use different stationary phases or only C18?
Answer- A second C18 column, although it had the same particle size and dimensions. This has been changed to “A different C18 column” in table S5.
Please, provide scans of vouchers of specimens as supporting material. If it is possible, provide also photos of the purchased plant material.
Answer- We have included the vouchers in the supplementary material, as requested, but, unfortunately, we did not photograph the commercial samples.
Read the whole manuscript conscientiously, please, and correct several typos and discrepancies.
Answer- The manuscript was revised.
Round 2
Reviewer 1 Report
The authors provide and answer all the comments. I agree with the comments of the authors y they have improved the manuscript, which has the quality to publish in the journal.
I thank the authors for considering my comments.